# Application of “Magnetic Anchors” to Align Collagen Fibres for Axonal Guidance

**DOI:** 10.3390/gels7040154

**Published:** 2021-09-27

**Authors:** Devindraan S/O Sirkkunan, Farina Muhamad, Belinda Pingguan-Murphy

**Affiliations:** Department of Biomedical Engineering, Faculty of Engineering, Universiti Malaya, Kuala Lumpur 50603, Malaysia; ezvn2066@gmail.com (D.S.); bpingguan@gmail.com (B.P.-M.)

**Keywords:** gold magnetic nanoparticles, collagen mimetic peptide, magnetic field alignment, “magnetic anchor” method, orientation of PC12 neurite extensions

## Abstract

The use of neural scaffolds with a highly defined microarchitecture, fabricated with standard techniques such as electrospinning and microfluidic spinning, requires surgery for their application to the site of injury. To circumvent the risk associated with aciurgy, new strategies for treatment are sought. This has led to an increase in the quantity of research into injectable hydrogels in recent years. However, little research has been conducted into controlling the building blocks within these injectable hydrogels to produce similar scaffolds with a highly defined microarchitecture. “Magnetic particle string” and biomimetic amphiphile self-assembly are some of the methods currently available to achieve this purpose. Here, we developed a “magnetic anchor” method to improve the orientation of collagen fibres within injectable 3D scaffolds. This procedure uses GMNP (gold magnetic nanoparticle) “anchors” capped with CMPs (collagen mimetic peptides) that “chain” them to collagen fibres. Through the application of a magnetic field during the gelling process, these collagen fibres are aligned accordingly. It was shown in this study that the application of CMP functionalised GMNPs in a magnetic field significantly improves the alignment of the collagen fibres, which, in turn, improves the orientation of PC12 neurites. The growth of these neurite extensions, which were shown to be significantly longer, was also improved. The PC12 cells grown in collagen scaffolds fabricated using the “magnetic anchor” method shows comparable cellular viability to that of the untreated collagen scaffolds. This capability of remote control of the alignment of fibres within injectable collagen scaffolds opens up new strategic avenues in the research for treating debilitating neural tissue pathologies.

## 1. Introduction

Neurological disorders (NDs) that result from diseases and injury usually involve the loss of neurons in various parts of the brain, spinal cord, and other parts of the all-encompassing peripheral nervous system. Despite the substantial decrease in mortality rates from stroke and communicable neurological diseases, the burden of ND has increased in the past 25 years due to the gradual increment of the aged population [1]. Hence, there is a need to prepare more efficient methods to provide adequate treatment for the increasing number of patients with ND.

The current treatment strategy for ND that involves neuronal loss, such as trauma to the spinal cord and peripheral nerves, is definitive surgical decompression and/or stabilisation [2]. The autologous peripheral nerve graft has also been used for treating traumatic injuries and other NDs such as Parkinson’s disease [2]. However, this procedure of stabilisation requires the transference of nerves from another part of the nervous system, as seen in the excision of sural nerves containing Schwann cells and their delivery into the Parkinson’s disease-affected substantia nigra [2]. The increment in the morbidity of patients during surgical procedures motivates research for other avenues in ND treatment technologies.

As an alternative to nerve transplantation, stem cell therapy provides a renewable source of auxiliary cells and tissues for a variety of NDs [3]. Bone marrow cell transplantation has been used to treat spinal cord injury, and has been shown to be a viable option for patients with complete spinal cord injury [4]. However, the research shows small improvements in the treatment of acute and sub-acute groups, but not in chronic groups [4]. This is due to the fact that stem cells require a scaffold and vector to improve their functionality [4,5]. 

Biomimetic nanofibrous scaffolds have been developed for neural tissue engineering in order to provide sustained growth factor/drug release or to support cell growth in situ [5,6]. The ability to fine tune the biochemical properties of the nano-fibres enables researchers to produce biomaterials that can mimic the extracellular matrix (ECM) of native tissues [7,8]. This potential, coupled with a high surface area to volume ratio and superior biocompatibility, provides a technique to reduce cell death or neuropathy due to non-physiological local stress [8,9]. Hence, many methods have been developed to fabricate these scaffolds according to the desired functionality and specifications.

Electrospinning is frequently used to fabricate scaffolds due to its ability to manipulate the developmental parameters, such as porosity, surface area, fibre diameter, and fibre alignment [10]. It is considered a standard technique for producing nano-fibres in the field of neural tissue engineering [11,12]. Another emerging technique to fabricate neural scaffolds is microfluidics [13,14]. This method does not require the application of a high temperature or voltage [15,16]. There are also other novel methods being developed to produce neural scaffolds, such as isoelectric focusing [17], wet spinning [18], and the thermal drawing process [19]; with each technique improving the scaffold attributes that can enhance and direct cellular growth. 

However, these scaffolds that are shaped and solidified in vitro require surgical insertion at the site of injury. A more immediate method of therapy would include on-site treatment to reduce scarring or accumulation of inhibitory proteins. Hence, in recent years, researchers have examined injectable hydrogels as a viable option for minimally invasive treatment of various injuries [20,21,22]. These hydrogels can be administered immediately after injury, and because surgery is not required to apply the scaffold, the morbidity from surgical trauma can be greatly reduced [23].

Despite their malleability and ease of application, injectable hydrogels do not have the defined microarchitecture that is usually found in in vitro patterned scaffolds [24,25,26]. This topology within scaffolds plays an important role in orientating cellular growth and building appropriate microstructures that could provide sustained tissue development [24,25,26]. Current advances in this new class of hydrogels employ nanoparticles and nanotubes to direct polymer fibres therein and, in effect, direct the alignment of cells [24,25,26].

Here, we report a new method to orientate the internal structure of injectable scaffolds made of collagen using collagen mimetic peptide (CMP)-functionalised gold magnetic nanoparticles (GMNPs). Gold nanoparticles have been previously used in enhancing the resolution of viewing collagen fibres, but CMPs used in conjunction with gold magnetic nanoparticles to improve the orientation of the collagen fibres is apparently novel [27]. The focus of this study is to introduce a new method to fabricate injectable scaffolds with improved collagen fibre orientation, and to investigate their effect on the directionality of PC12 cells’ axonal growth.

## 2. Results and Discussion

Plain collagen scaffolds as shown in Figure 1A was fabricated using standard collagen preparation methods. In this study, the efficiency of the “magnetic anchor” method was compared to the “magnetic particle string” method, which was described in a previous study with similar goals of improving axonal alignment [26]. In that study, magnetic nanoparticles formed strings that aggregated along the magnetic field and orientated the collagen fibres therein [26]. They also acted as a guide to direct cellular axonal growth [26]. However, in our study, GMNPs were used instead of magnetic nanoparticles to form magnetic particle strings, due to their multifunctional capability of surface modification (the attachment of CMPs on the exposed gold nanoparticle) and high responsiveness to a magnetic field (iron oxide nanoparticle core). The mechanism of aligning the collagen fibres remains the same, namely, the GMNP particle strings force the collagen fibres into orientating along the magnetic fields (Figure 1B). In contrast, the “magnetic anchor” method was made possible with the incorporation of CMPs into GMNPs. The CMP “chains” bind the GMNP “anchors” firmly to the collagen fibres, which then “dock” along the magnetic field lines during collagen solidification. (Figure 1C). Hence, this application of CMP-functionalised GMNPs enhances the collagen fibres’ alignment when the magnetic field is applied. The phase contrast microscope image in Figure 1A shows a pure collagen scaffold, which is very clear compared to Figure 1B,C, which contains only GMNPs or CMP-functionalised GMNPs, respectively. The magnetic particle strings are clearly visible in Figure 1B, and nearly absent in Figure 1C, which shows a more even distribution of the CMP-functionalised GMNPs in the collagen scaffolds. However, there are aggregations seen in Figure 1C, which may be due to GMNPs that were not surface-modified with CMPs, and therefore, aggregated along the magnetic field line.

### 2.1. UV Vis

Functionalisation of GMNPs with Cysteine-CMP (Cys-CMP) was assessed using UV-Vis spectroscopy. Cys-CMP forms thiol bonds with gold nanoparticles on the surface of GMNPs. This increases the size and changes the surface plasmon resonance of the nanoparticles, and shows a red shift in the absorption spectra [28,29]. This is seen in Figure 2, where the unmodified GMNPs with a peak of 958 nm red shifted to 965 nm after functionalisation with Cys-CMP.

### 2.2. SEM Image and Analysis

SEM images of the scaffolds were analysed using the directionality analysis utility software from FIJI [26,30]. The normalised pixel intensity along a specific radius was represented in a Fast Fourier Transform (FFT) analysis plot, which compares the fibre alignment arbitrary unit (a.u) of the collagen scaffolds. The red arrows seen in SEM images Figure 3A,C,E point to PBS crystals that formed during the freeze-drying process. Hence, areas with minimum PBS crystals, denoted by the yellow dotted squares, were chosen for analysis. The FFT analysis plot in Figure 3F shows a significantly higher degree of fibre alignment in scaffolds fabricated with the “magnetic anchor” method, with a singular highest peak value of 0.0451 a.u. at 76° (indicated with a blue arrow in Figure 3F) compared to the other samples, which have an average peak of 0.024 ± 0.002 a.u. at various corresponding degrees. 

A comparison FFT analysis plot of three different scaffold samples (N = 3) fabricated with their respective methods was also performed to determine the consistency of the results obtained, and is shown in Figure 4A–E. The FFT analysis plot of scaffold samples fabricated with the “magnetic anchor” method shown in Figure 4E reports an average peak value of 0.044 ± 0.003 a.u., which was the highest peak value of all the scaffold types compared. Furthermore, the plot for all of the scaffold samples fabricated with the “magnetic anchor” method has a singular and defined highest peak at 74°, 36°, and 24° for scaffold sample 1, scaffold sample 2, and scaffold sample 3, respectively (indicated with blue arrows in Figure 4E), which shows that the alignment of the collagen fibres is significantly focused along this angle. In the FFT analysis plot shown in Figure 4C, scaffolds fabricated with the “magnetic particle string” method show an average peak value of 0.027 ± 0.005 a.u., which is nearly half the value obtained using the “magnetic anchor” method. The peak indicated by the green arrow in Figure 4C points to the singular highest peak of 0.032 a.u. at −78° for scaffold sample 2, which shows greater alignment along this angle. However, this type of singular, high peak could not be seen in the other scaffold samples fabricated using the “magnetic particle sting” method, which shows that this method was less consistent in aligning collagen fibres. The FFT plots of scaffolds from other fabrication methods that do not involve the use of a magnetic field, shown in Figure 4A,B,D have an average peak value of 0.023 ± 0.004 a.u. with no singular, defined peak at any angles to denote any significant alignment of the collagen fibres. The singular, defined peaks in the FFT plots of scaffolds fabricated with the “magnetic particle string” and “magnetic anchor” methods show that the natural aggregation of the nanoparticles is not the cause of alignment; rather, the cause is a combination of CMP-attached nanoparticles and the application of a magnetic field. The higher peak values obtained in all three scaffold samples fabricated using the “magnetic anchor” method indicates this new strategy to align collagen fibres is both exceptional and reliable.

### 2.3. Phalloidin/DAPI Confocal Image and Analysis

Figure 5A–E shows differentiated PC12 cells within the various types of scaffolds compared in this study. There are no visible neurite extensions in plain collagen scaffolds, or GMNP-incorporated collagen scaffolds without magnetic field treatment, as shown in Figure 5A,B respectively. Therefore, only the other three scaffold types (scaffolds fabricated with the “magnetic particle string” method, CMP-functionalised GMNP collagen scaffolds without magnetic field treatment, and scaffolds fabricated with the “magnetic anchor” method) were chosen for axonal alignment and neurite length comparison. Figure 5F shows the method in which the differentiated PC12 cells were measured. The method of deriving the angle of deviation is shown in Figure 5G. Using this standardised method, the precision of neurite orientation could be determined, which ultimately reflects the efficacy of the techniques used to control collagen fibre alignment. The exact angle of deviation of the neurites from the mean is shown in Appendix A. For this analysis, a normal distribution curve was used to represent the precision of axonal alignment. The data of the probability density function used to plot the normal distribution curve are shown in Appendix A. The significance study of the correlation between the compared groups performed using ANOVA and the Tukey post hoc test is shown in Appendix A. As shown in Figure 5H, scaffolds fabricated with the “magnetic anchor” method showed a narrower bell curve compared to the other types of scaffold. This is due to its smaller standard deviation (SD) of 26.161° and coefficient of variation (COV) of 0.2031, compared to scaffolds fabricated with the “magnetic particle string” method, which had a SD of 52.215° and COV of 0.6215, and the CMP-functionalised GMNP collagen scaffolds without magnetic field treatment, which had a SD of 59.652° and COV of 0.5598. 

The modified standard normal distribution curve shown in Figure 6A indicates that the differentiated PC12 cells in scaffolds fabricated with the “magnetic anchor” method have a higher probability of axons aligning within 30° of the mean angle of alignment. This can also be seen in the histogram shown in Figure 6E, which has a chi-square value of 11.8351, df of 3, *p* = 0.00797, and shows a higher number of axons aligning within ±30°of the mean. Specifically, the growth orientation analysis shows that 41% of the neurites exhibited alignment within 15° (Δθ < 15°), and 86% exhibited alignment within 30° (Δθ < 30°). In contrast, neurites on the other two scaffold types show a more dispersed orientation from the mean. The histogram of the differentiated PC12 cells’ axonal orientation within CMP-functionalised GMNP collagen scaffolds without magnetic field treatment shows that only 11% of the neurites exhibited Δθ < ±15°, and 17% exhibited Δθ < ±30°, as seen in Figure 6D (chi-square 13.3878, df of 3, and *p* = 0.003868). The axonal orientation of the differentiated PC12 cells within scaffolds fabricated with the “magnetic particle string” method show a higher value, with 22% of the neurites exhibiting Δθ < ±15°, and 38% exhibiting Δθ < ±30°, as seen in Figure 6C (chi-square 1.8413, df of 3, and *p* = 0.605989). However, according to the chi-square value, the alignment results are not significant within 60° (*p* > 0.05). The axons of differentiated PC12 cells cultured within scaffolds fabricated using the “magnetic anchor” method also exhibit a significant increase in length (67.48 μm) compared to axons of cells within scaffolds fabricated with the “magnetic particle string” method (60.78 μm), as shown in Figure 6B. These results indicate that the use of CMP-functionalised GMNPs and magnetic field application during the collagen’s gelling process are necessary to produce a highly aligned internal microarchitecture, which leads to efficient orientation of PC12 neurites cultured therein. 

### 2.4. LIVE/DEAD Confocal Image and Analysis

To further examine the capability of these scaffolds as a nerve regeneration platform, LIVE/DEAD tests were performed to assess the PC12 cell’s viability cultured within. The confocal images of the scaffolds are shown in Figure 7A–E. Using Image J to tabulate the live and dead cells as stained by calcien AM and heterohomodimer-1, respectively, the percentages of live cells were determined. The histogram in Figure 7F shows that there was not a significant difference in the cellular viability between the different types of scaffold. ANOVA revealed that there were no significant differences when the groups were compared with one another, which is consistent with previous studies undertaken using nanoparticles [26,31]. The significance study of the correlation between the compared groups performed using ANOVA and the Tukey post hoc test is shown in Appendix A. Hence, it was concluded that the PC12 cells grown in these scaffolds were not adversely affected by the use of GMNPs, CMPs, or the magnetic field.

### 2.5. Discussion

Numerous technologies to align fibres have been developed in recent decades, which mainly involve the application of magnetic nanoparticles or anisogel/nanoparticles to direct these fibres [24,25,26]. In a study by Antman-Passig and Orit Shefi, 2016, the magnetic nanoparticles in collagen gels form “magnetic particle strings” in a magnetic field, and this string subsequently pushes the collagen fibres to orientate during their gelling process [26]. This method provides a direct and simple means to improve the alignment of fibres [26].

In the current study, we introduced a different mechanism to align collagen fibres. Our approach is characterised by a similar degree of simplicity but has greater potential for fibre alignment and axonal growth control. In the presented mechanism, GMNPs were used instead of regular magnetic nanoparticles, and attached to collagen fibres using CMPs. The collagen fibres were then pulled to orientate along the magnetic field. Due to the action of these GMNPs docking the collagen fibres along the magnetic field, they were termed “magnetic anchors”. Hence, it is necessary to determine the extent of improvements resulting from this method in the context of current fibre alignment research. It should be noted that the original research by Antman-Passig and Orit Shefi, 2016, using the “magnetic particle string” method utilised a significantly higher concentration of magnetic nanoparticles (0.05 mg per 100 μL collagen solution), whereas in our study, the quantity of nanoparticles used was considerably less (≈0.005 mg per 100 μL collagen solution). Therefore, the quantity of nanoparticles used for both fabrication methods was standardised to 0.005 mg per 100 μL collagen solution.

As seen in the SEM images and FFT plot analysis, the collagen fibres show remarkable alignment when the “magnetic anchor” method was utilised for scaffold fabrication, compared to the “magnetic particle string” method and plain untreated collagen scaffolds. The results obtained for the “magnetic particle string” method were also not consistent, because one scaffold sample fabricated with this method showed alignment, whereas the other two samples did not. In contrast, all three scaffold samples fabricated with the “magnetic anchor” method consistently showed exceptional alignment, which surpasses all the samples compared in this study. The impact of this fibre alignment could be clearly seen in the efficacy of PC12 cells’ neurite orientation. The neurites of PC12 cells grown in scaffolds fabricated using the “magnetic anchor” method mostly (86%) aligned within ±30° of the mean. By comparison, only 38% of the neurites of PC12 cells within the “magnetic particle string” scaffolds attained an alignment within ±30° of the mean.

Another characteristic of differentiated PC12 cells that was compared in this study was neurite length. Neurite length in PC12 cells has also previously been used to indicate the extent of differentiation, where longer neurites have been shown to correspond with greater release of synapsin-1 and gap-43, which are markers of differentiated PC12 cells [32,33]. In the current study, the longest neurites were from differentiated PC12 cells cultured in scaffolds fabricated using the “magnetic anchor” method. This also shows the method’s capability of producing scaffolds that can enhance PC12 cells’ differentiation. An interesting point to note is that the neurites of differentiated PC12 cells cultured in scaffolds fabricated with the “magnetic particle string” method were even shorter than the neurites of PC12 cells cultured in the conventionally fabricated collagen scaffolds. This may be due to the force applied during scaffold formation, in which the formation of a “magnetic particle string” projects a higher, localised force, whereas the “magnetic anchors” create a more even, albeit lower force across the scaffold. However, more research must be undertaken to elucidate the exact mechanism that led to this phenomenon.

The final characterisation to determine the PC12 cell viability within these specially fabricated scaffolds was performed using the live/dead test. The results revealed that the percentage of live cells was higher in the “magnetic anchor” scaffolds. However, ANOVA and the Tukey post hoc test performed on the data showed no significant difference between all the groups compared (*p* > 0.1). These results concur with cell viability tests performed in other studies that utilise nanoparticles of micron size [26,30]. Therefore, the “magnetic anchor” method reported in this paper was shown to guide the growth of PC12 neurites and improve upon the cells’ differentiation process without eliciting any cell-damaging effects.

In summary, a novel method was developed to improve the internal alignment of 3D scaffolds made from collagen hydrogels by utilising the “magnetic anchor” method, which incorporates CMP-functionalised GMNPs attached to collagen fibres that “dock” along the magnetic field of a magnet during the hydrogel’s gelling process. As seen in the SEM images and FFT plot analysis, the collagen fibres showed excellent alignment when this method was utilised. Furthermore, the study of PC12 cells’ differentiation showed greater precision of axonal orientation and longer extensions of neurites. The cellular viability was also similar between the scaffolds fabricated with the “magnetic anchor” method and plain collagen gels, which shows that this method of fabrication is safe for PC12 cell culture and differentiation.

## 3. Conclusions

In recent years, research findings relating to scaffold technology have emphasised the importance of micro- and nano-internal architectures to manipulate cellular growth. Numerous standard and popular methods, such as electro- and microfluidic spinning, have been used to achieve this. Although highly defined structurally, the application of these scaffolds requires surgery, which has prompted researchers to search for an alternative approach. Injectable hydrogels are the natural solution to this problem. However, the patterning of the scaffold’s internal “framework” must happen automatically or via remote control after injection into the affected site. This remains a therapeutic challenge when utilising these injectable gels in an efficient manner. In this study, a “magnetic anchor” method was proposed to maximise the efficiency of collagen fibre alignment during the solidification process, using CMP-functionalised GMNPs in a magnetic field. This will allow the injection of these collagen gels into the site of injury and, with the application of an external magnetic field, the scaffoldings’ internal fibres will orientate for cellular guidance and appropriate healing. Our study, in addition to other recent medical technologies involving GMNPs and magnetic fields, creates more strategies for remotely controlling scaffolds’ functionality for their application in the field of tissue engineering.

## 4. Materials and Methods

The materials used in this study were as follows: sodium hydroxide, 1N (NaOH) (R&M Chemicals), diethylene glycol (DEG) (R&M Chemicals, Subang, Selangor, Malaysia), polyvinyl pyrolidone (PVP) (R&M Chemicals, Subang, Selangor, Malaysia), iron (II,III) oxide (powder) (Sigma-Aldrich, Bandar Sunway, Petaling Jaya, Malaysia), silica (SiO_2_) powder (Sigma-Aldrich Bandar Sunway, Petaling Jaya, Malaysia), isopropanol (Sigma-Aldrich Bandar Sunway, Petaling Jaya, Malaysia), ethanol, 95% technical grade (SYSTERM, Shah Alam, Selangor, Malaysia), 3-aminopropyl-1 triethoxysilane (APTS) (Sigma-Aldrich Bandar Sunway, Petaling Jaya, Malaysia), trisodium citrate (dihydrate)(R&M Chemicals, Subang, Selangor, Malaysia), gold (III) chloride hydrate (~50% Au basis) (Sigma-Aldrich Bandar Sunway, Petaling Jaya, Malaysia), Cysteine-Collagen mimetic peptide [Cys-(ProHypGly)_7_] (Apical Scientific sdn. Bhd, Seri Kembangan, Selangor, Malaysia) RPMI 1640 medium with 2.5 mM L-glutamine (with phenol red) (Gibco, Western St, Suite C, Amarillo, TX, USA), Fetal bovine serum (FBS) (Gibco, Western St, Suite C, Amarillo, TX, USA), Heat Inactivated Horse Serum (Gibco), Penicillin-Streptomycin (10,000 U/mL) (Sigma-Aldrich Bandar Sunway, Petaling Jaya, Malaysia), NGF-β from rat (Sigma-Aldrich Bandar Sunway, Petaling Jaya, Malaysia), Collagen I, rat tail (3 mg/mL) (Gibco, Western St, Suite C, Amarillo, TX, USA), N50 Neodymium Cuboid Magnets (30 × 10 × 3 mm), PC12 rat pheochromocytoma cells purchased from AddexBio Morena Blvd, Ste 1403 San Diego, CA 92117, Alexa Fluor 488 (Life Technologies, Western St, Suite C, Amarillo, TX, USA), Slowfade gold antifade reagent with DAPI (Thermofisher Scientific, Western St, Suite C, Amarillo, TX, USA), and Live/dead imaging kit (Thermofisher Scientific, Western St, Suite C, Amarillo, TX, USA).

### 4.1. Fabrication of Gold Magnetic Nanoparticles (GMNPs)

The fabrication process of GMNPs used a “one pot” polyol synthesis method, which is a modified version of the high temperature hydrolysis process that was developed previously [34,35]. Basically, four different solutions were prepared before the GMNP synthesis process. The first solution consisted of 50 mmol sodium hydroxide (NaOH) in 20 mL DEG; the second contained 12.5 g silica powder in 20 mL of 70% ethanol heated at 80 °C for an hour; the third solution was a mixture of 100 µL of APTS and 25 mL isopropanol heated at 80 °C for an hour; and the fourth solution contained a mixture of 25 mg trisodium citrate in 20 mL of 70% ethanol, heated to 80 °C followed by the addition of 5 mg gold (III) nitrate hydrate. To begin the “one pot” polyol synthesis, 20 mg PVP and 20 mg iron (II,III) oxide (Fe_3_O_4_) was added to 17 mL DEG and heated at 220 °C for 30 min. A quantity of 1.75 mL of the first solution was then added and heated at 220 °C. After an hour, the second solution was added, and this mixture was heated at a lower temperature of 150 °C for 1.5 h. Then, a pipette was used to add the third and the fourth solutions in a dropwise manner, and this final mixture was heated to 80 °C for an hour to produce the GMNPs. These GMNPs had a diameter of around 210 nm.

### 4.2. Functionalisation of GMNPs with Collagen Mimetic Peptides (CMPs)

A quantity of 5 mL of the GMNP suspension was pipetted into a 15 mL falcon tube for multiple cycles of cleaning with distilled water (dH_2_O). This was performed by removing the supernatant after the GMNPs settled and adding 10 mL of dH_2_O. The dH_2_O and GMNPs were mixed thoroughly and centrifuged at 1500 RPM for 5 min. The supernatant was then removed and another 10 mL of dH_2_O was added. This step of cleaning with dH_2_O and centrifugation was repeated 10 to 15 times, until a clear supernatant was seen after the nanoparticles precipitated at the bottom of the falcon tube. The weight of the nanoparticles was measured using an electromagnetic balance. The final volume of the suspension was adjusted to 11 mg/mL, and the content was emptied into a glass vial. The suspension was then placed in an autoclave for sterilisation and left to cool at room temperature for a day. The GMNPs in dH_2_O were then ready for scaffold fabrication. To produce CMP-functionalised GMNPs, 1.25 mg of CMP was added to 1 ml of the sterile GMNP in dH_2_O suspension within a Class II safety cabinet. This suspension was left at room temperature for 24 h. The CMP-functionalised GMNPs in dH_2_O were then ready for characterisation (UV-Vis spectroscopy) and scaffold fabrication.

### 4.3. Fabrication of Collagen Scaffolds (Collagen Gelling Protocol)

Plain collagen scaffolds were prepared as detailed in Gibco’s collagen gelling protocol, where the volumes of solutions used were as follows: Collagen (833 µL), 10 × PBS (100 µL), 1N NaOH (21 µL), and dH_2_O (46 µL). The concentration of collagen used to produce these scaffolds was maintained at 3 mg/mL. Quantities of 10 × PBS, 1N NaOH, and dH_2_O were added first to an Eppendorf tube before pipetting in the collagen solution. The mixture was agitated gently before being placed on to the mould. The moulds (1.7 cm in diameter) were capable of holding 250 mL collagen mixture. The gelling process took roughly 10 min at room temperature in a class II safety cabinet. The moulds were then transferred into a 37 °C, 5% CO_2_ incubator for another 10 min to facilitate gelling. These scaffolds were then placed in a petri dish containing dH_2_O.

### 4.4. Fabrication of Scaffolds Using the “Magnetic String” Method and “Magnetic Anchor” Method

Due to natural aggregation of nanoparticles in collagen gels, it was necessary to determine if efficacy of collagen fibre alignment was the result of this aggregation or the application of a magnetic field. Hence, there were five types of scaffold prepared for this study: (1) plain collagen scaffolds; (2) GMNP-incorporated collagen scaffolds without magnetic field treatment; (3) GMNP-incorporated collagen scaffolds solidified in a magnetic field (scaffolds fabricated with the “magnetic particle string” method) shown in Figure 1C; (4) CMP-functionalised GMNP collagen scaffolds without magnetic field treatment; and (5) CMP-functionalised GMNP collagen scaffolds solidified in a magnetic field (scaffolds fabricated with the “magnetic anchor” method) shown in Figure 1B. To fabricate scaffolds containing either GMNPs or CMP-functionalised GMNPs, the 46 µL dH_2_O used in the collagen gelling protocol was replaced with 46 µL of these nanoparticles’ suspension in dH_2_O that was prepared earlier. To produce scaffolds using the “magnetic particle string” and “magnetic anchor” methods, the collagen gels incorporated with either type of nanoparticles were pipetted onto the moulds and placed in a magnetic field established as shown in Figure 8A. After the initial gelling period of 5 min at room temperature in the magnetic field established within a class II biosafety cabinet, the mould was transferred into a 37 °C, 5% CO_2_ incubator. The collagen scaffolds were then removed from the mould and placed in a petri dish containing distilled water, and were then ready for characterisation. The final concentration of GMNPs or CMP-functionalised GMNPs in collagen was 0.5 mg/mL, which follows the same concentration used in a previous study by Antman-Passig and Shefi [26], detailing the “magnetic particle string” method. Therefore, an initial concentration of 11 mg/mL GMNPs in dH_2_O was prepared for this purpose. The magnetic field of the N50 magnets used in the set-up shown in Figure 8A had a strength of 126 gauss at the circumference of the mould and a strength of 54 gauss at the centre of the mould (calculated using the K&J magnetic field calculator—https://www.kjmagnetics.com/calculator.asp-accessed on 16 July 2019). The summary of all of the fabrication processes compared in this study is shown in Appendix A.

### 4.5. PC12 Cell Culture in Collagen Scaffolds

PC12 rat pheochromocytoma cells were cultured in a T25 flask containing RPMI-1640 medium supplemented with 10% heat inactivated horse serum (HS), 5% foetal bovine serum (FBS), and 1% penicillin streptomycin, in a 37 °C, 5% CO_2_ incubator. Cells at passage 2 were used for this study. The final cell suspension was adjusted to 1 × 10^6^ cells per ml. PC12 cells were seeded in each of the five types of scaffold prepared for this study. For plain collagen scaffolds, 46 μL of cell suspension was used to replace the dH_2_O in the collagen gelling protocol. To prepare the cell solution for incorporation into the other types of scaffold, 46 µL of the nanoparticles in dH_2_O suspension was pipetted into an Eppendorf tube and spun at 1500 RPM for 5 min. The supernatant was then removed and 46 µL of the cell suspension was pipetted into the Eppendorf tube, giving a solution of approximately 46,000 PC12 cells and 500 µg nanoparticles in 46 µL complete RPMI-1640 culture medium. This suspension then replaced the dH_2_O in the collagen gelling protocol described above. This mixture was agitated carefully with a pipette and placed onto the mould. To produce scaffolds using the “magnetic particle string” and “magnetic anchor” methods, the collagen gels incorporated with either type of nanoparticles and PC12 cells were pipetted onto the mould and placed in a magnetic field set up as shown in Figure 1A. The gels were allowed to solidify at room temperature for 10 min, before transferring the mould from the magnetic field set-up, into a 37 °C, 5% CO_2_ incubator. The collagen scaffolds were then transferred into petri dishes containing complete RPMI-1640 culture medium and placed back into the 37 °C, 5% CO_2_ incubator for 24 h. The culture medium was then replaced with a differentiation-inducing medium made up of RPMI-1640 medium containing 1% HS, 1% penicillin streptomycin, and 50 ng/mL rat NGF-β. This differentiation medium was changed every two days, and characterisation (Phalloidin/DAPI staining and live/dead staining) was performed after the second treatment was completed (after 5 days from the initial seeding). The summary of all of the fabrication processes compared in this study is shown in Appendix A.

### 4.6. Characterisations and Analysis

#### 4.6.1. UV-Vis Spectroscopy

UV-Vis was recorded 24 h after the attachment of CMP to the GMNPs using a single monochromator UV-2600i Shimadzu in the range of 300–1000 nm at a resolution of 0.1 nm.

#### 4.6.2. Scanning Electron Microscopy (SEM)

Three different scaffold samples (N = 3) were fabricated with their respective methods to determine the alignment consistency of each method. A quantity of 10% formalin was used to fix the collagen scaffolds, which were then left at room temperature for 1 h in a class II biosafety cabinet. Subsequently, the scaffolds were washed repeatedly with PBS and subjected to serial dehydration with ethanol of increasing concentration, progressing from 50% to 70%, 80%, 90%, and finally 100% ethanol. The scaffolds were then freeze dried overnight. These freeze-dried scaffolds were then placed on aluminium stubs, sputter coated with gold, and viewed with a ZEISS Gemini SEM.

#### 4.6.3. Confocal Microscopy (Phalloidin and DAPI Staining)

The scaffolds were rinsed twice with PBS before fixing with 10% formalin for 10 min at room temperature. They were then rinsed again with PBS, two or more times, before applying 0.1% Triton X-100 prepared in PBS, and left for 5 min at room temperature. The PBS rinsing process was repeated two more times. A quantity of 1% bovine serum albumin (BSA) in PBS was then added to the scaffolds and they were incubated in a 37 °C, 5% CO_2_ incubator for 30 min to reduce nonspecific background staining. To prepare the phalloidin stain (Life technologies), 5 µL of the methanolic stock solution (6.6 μM) was diluted in 200 µL PBS for a final concentration of 1.165 μM. A quantity of 10 μL of this phalloidin staining solution was then pipetted onto the scaffolds and they were left for 20 min at room temperature in a Class II safety cabinet. The scaffolds were then rinsed with PBS twice before adding Slowfade gold antifade reagent with DAPI. The scaffolds were left at room temperature for 5 min in a Class II safety cabinet, before rinsing with PBS twice and viewed with a confocal microscope (Leica TCS SP5). Phalloidin stained cells were examined at green fluorescence (ex/em 495 nm/519 nm) and DAPI stained cells at blue fluorescence (ex/em 345 nm/455 nm).

#### 4.6.4. LIVE/DEAD Assay

To prepare the working solution of LIVE/DEAD assay, 20 µL of 2 mM of ethidium homodimer-1 stock solution was added to 10mL PBS. After properly mixing the solution, 5 µL of 4 mM calcien AM stock solution was added. This produced a working solution of 2 µM calcein AM and 4µM ethidium homodimer-1. After removing the culture medium from the petri dishes and washing the scaffolds with PBS twice, 50 µL of the live/dead working solution was added onto the scaffolds, and they were incubated for 45 min at room temperature. After the incubation period, the scaffolds were cleaned with PBS and viewed with a confocal microscope (Leica TCS SP5). Live cells were examined at green fluorescence (ex/em 494 nm/517 nm) and dead cells at red florescence (ex/em 528 nm/617 nm).

#### 4.6.5. Image and Statistical Analysis

Quantitative evaluation of the SEM images was performed using two-dimensional (2D) fast Fourier transform (FFT), calculated through the directionality plugin in the FIJI software [26,36]. A total of 15 collagen scaffolds were analysed for the SEM characterisation. Three different areas on the collagen scaffolds were chosen at random for the SEM images. The morphometric parameter of neuron-like differentiated PC12 cells from confocal microscopy images stained with phalloidin/DAPI were also measured using the FIJI open-source image analysis software. The axon-like structures were traced using the Simple Neurite Tracer plugin within the FIJI software to determine both the directionality of the axons and their length [26]. Measurement of directionality was performed by determining the difference in the angle of deflection between the individual axon and the average angle of alignment of the total axons within the scaffold, as shown in Figure 5F,G and Figure 8D. Hence, a normal distribution was used to represent the data, because it conveyed the precision of the dataset. The more bell-shaped the curve, the more precise the dataset (closer to the mean value); in this study, this indicated greater alignment of the axons within the scaffolds, and vice versa. The neurite length comparison between treatments was represented with a bar charts. For the phalloidin/DAPI staining, a total of 125 cells were analysed. Quantification of cell number for live/dead confocal images was performed using Image J analysis software, as shown in the study by Spaepen, P et al. [37]. A total of 10 collagen scaffolds were used for the live/dead assay, and approximately 4109 cells were tabulated and analysed using the software. A one-way analysis of variance (ANOVA) with the Tukey HSD (honest significance difference) post hoc test was used to analyse the data obtained for neural length. The statistical significance of the angle of deviation values binned at 15° was determined with the chi-squared test.

## Figures and Tables

**Figure 1 gels-07-00154-f001:**
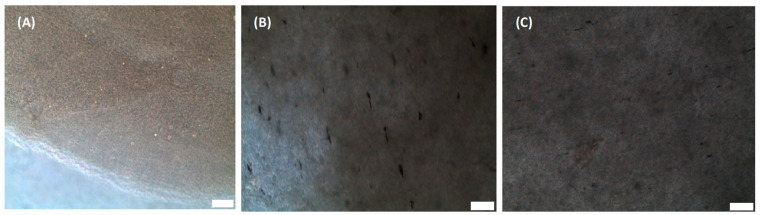
Phase contrast microscope image of (**A**) plain collagen scaffolds, (**B**) phase contrast microscope image of scaffolds fabricated using the “magnetic particle string” method, and (**C**) phase contrast microscope image of scaffolds fabricated using the “magnetic anchor” method. Scale bar = 50 µm.

**Figure 2 gels-07-00154-f002:**
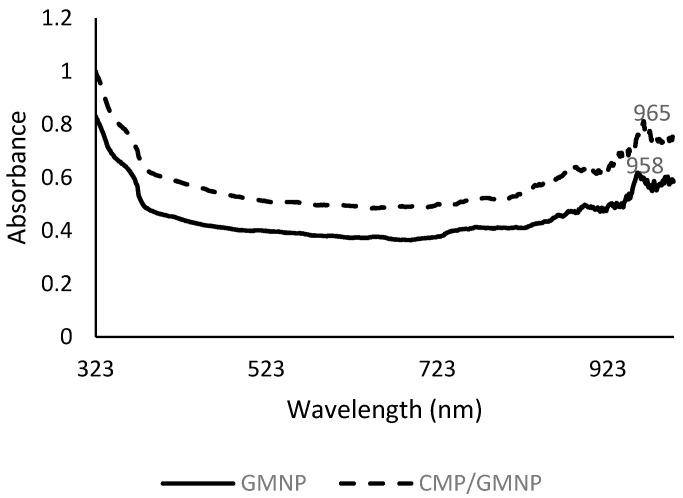
Comparative UV-Vis spectra that shows unmodified GMNPs and the surface-modified CMPs/GMNPs.

**Figure 3 gels-07-00154-f003:**
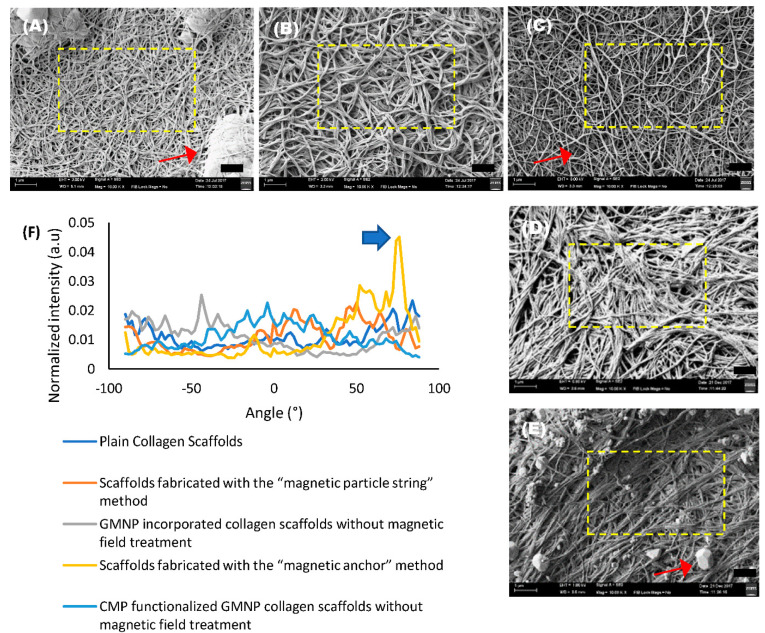
SEM images of collagen scaffolds and alignment analysis: (**A**) SEM image of plain collagen scaffold; (**B**) SEM image of GMNP-incorporated collagen scaffold without magnetic field treatment; (**C**) SEM image of scaffold fabricated with the “magnetic particle string” method; (**D**) SEM image of CMP-functionalised GMNP collagen scaffold without magnetic field treatment; (**E**) SEM image of scaffold fabricated with the “magnetic anchor” method; (**F**) FFT spectrum analysis plot of the areas depicted by the yellow dotted box in (**A**–**E**). Scale bar = 1 µm.

**Figure 4 gels-07-00154-f004:**
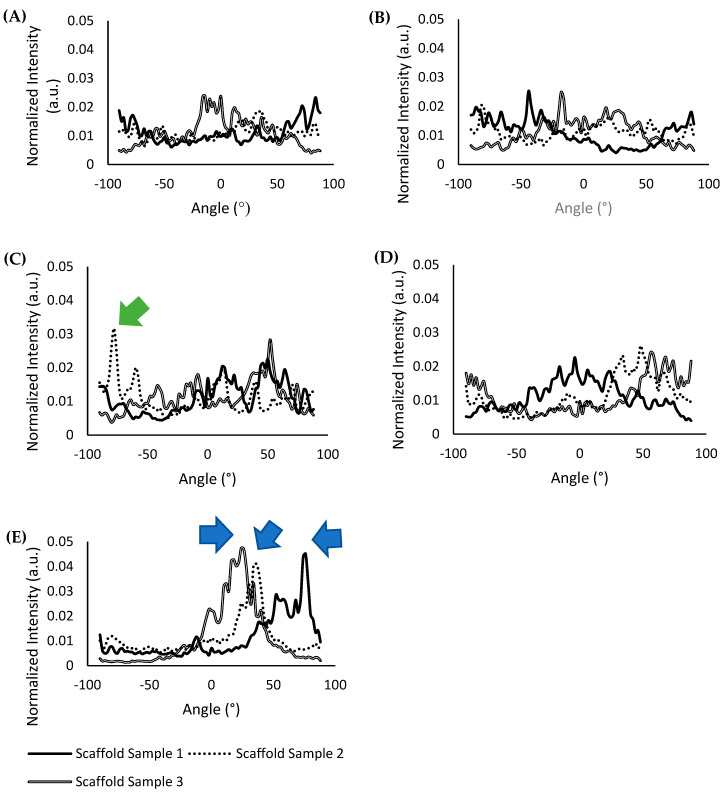
FFT analysis plot of the scaffold samples fabricated using their corresponding methods: (**A**) plain collagen scaffolds; (**B**) GMNP-incorporated collagen scaffolds without magnetic field treatment; (**C**) scaffolds fabricated with the “magnetic particle string” method; (**D**) CMP-functionalised GMNP collagen scaffolds without magnetic field treatment; (**E**) scaffolds fabricated with the “magnetic anchor” method.

**Figure 5 gels-07-00154-f005:**
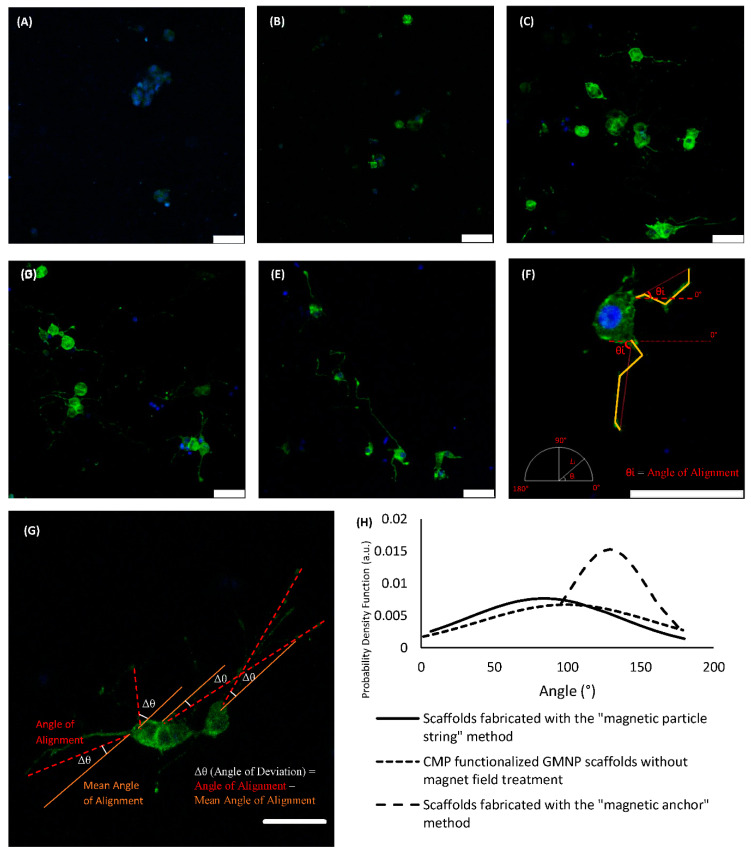
Confocal images of Phalloidin/DAPI staining of PC12 cells within plain collagen scaffolds (**A**), GMNP-incorporated collagen scaffolds without magnetic field treatment (**B**), scaffolds fabricated with the “magnetic particle string” method (**C**), CMP-functionalised GMNP collagen scaffolds without magnetic field treatment (**D**), and scaffolds fabricated with the “magnetic anchor” method (**E**). Image (**F**) shows the method in which the differentiated PC12 cells were measured; the yellow lines indicate the axonal length and red lines indicate the plane of alignment. θ represents the angle of alignment calculated from the horizontal plane (0°). (**G**) shows the method to derive the angles used in this analysis. (**H**) shows the alignment of axons represented by normal distribution curves for treatments that showed axonal growth. Scale bar = 50 µm.

**Figure 6 gels-07-00154-f006:**
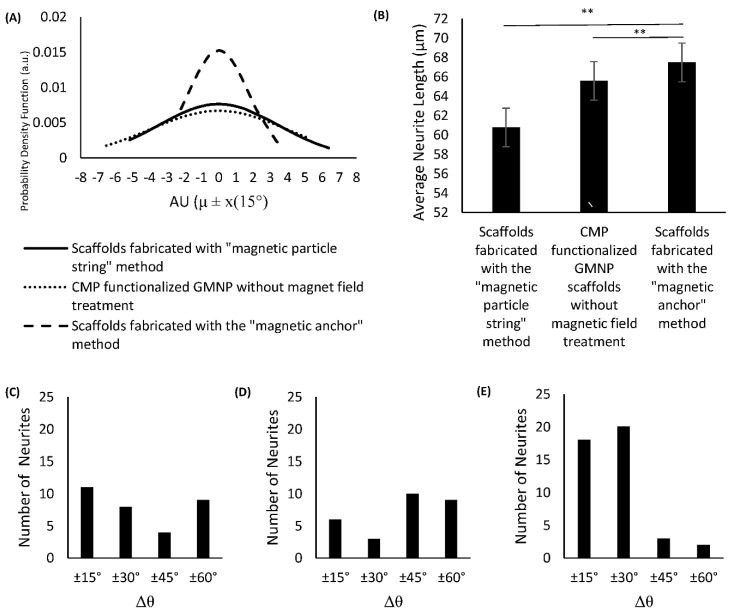
Image (**A**) shows a modified standard normal distribution curve (with μ = 0 and σ = 15° for all samples) of axons’ angle of deviation from the mean angle of alignment. Image (**B**) shows a comparative bar chart representing neurite length of differentiated PC12 cells for treatments that exhibited neurite extensions. (**C**–**E**) show histograms of PC12 neurite angle of deviation from the mean value, binned at 15°, of: (**C**) scaffolds fabricated with the “magnetic particle string” method (chi-square 1.8413, df 3, and *p* = 0.605989), (**D**) CMP-functionalised GMNP collagen scaffolds without magnetic field treatment (chi-squared 13.3878, df 3, and *p* = 0.003868), and (**E**) scaffolds fabricated with the “magnetic anchor” method (chi-squared 11.8351, df 3, and *p* = 0.00797). ** *p* < 0.01.

**Figure 7 gels-07-00154-f007:**
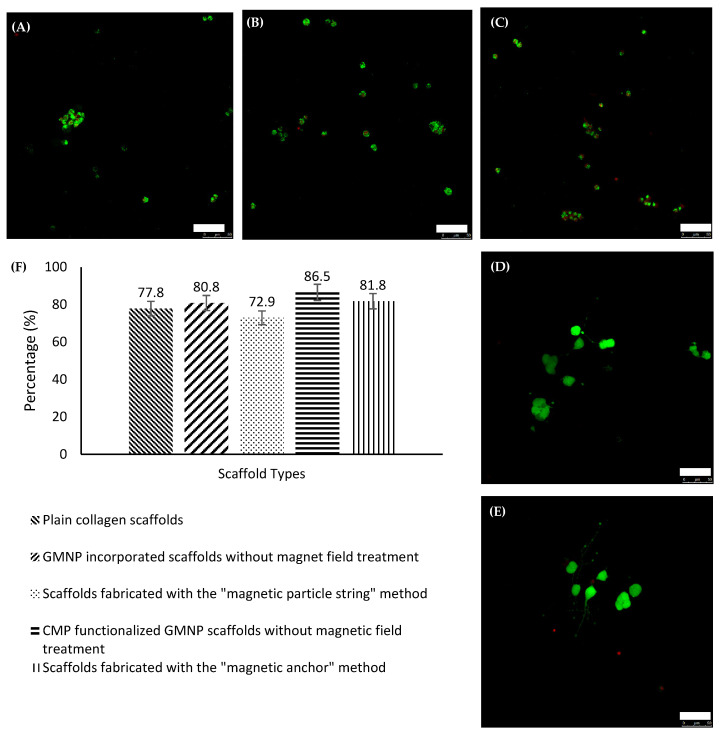
Confocal images of live/dead staining for PC12 cells within (**A**) plain collagen scaffolds, (**B**) GMNP-incorporated collagen scaffolds without magnetic field treatment, (**C**) scaffolds fabricated with the “magnetic particle string” method, (**D**) CMP-functionalised GMNP collagen scaffolds without magnetic field treatment, and (**E**) scaffolds fabricated with the “magnetic anchor” method. (**F**) shows the bar chart of the percentage of live cells within hydrogels of every treatment group. Scale bar = 50 µm.

**Figure 8 gels-07-00154-f008:**
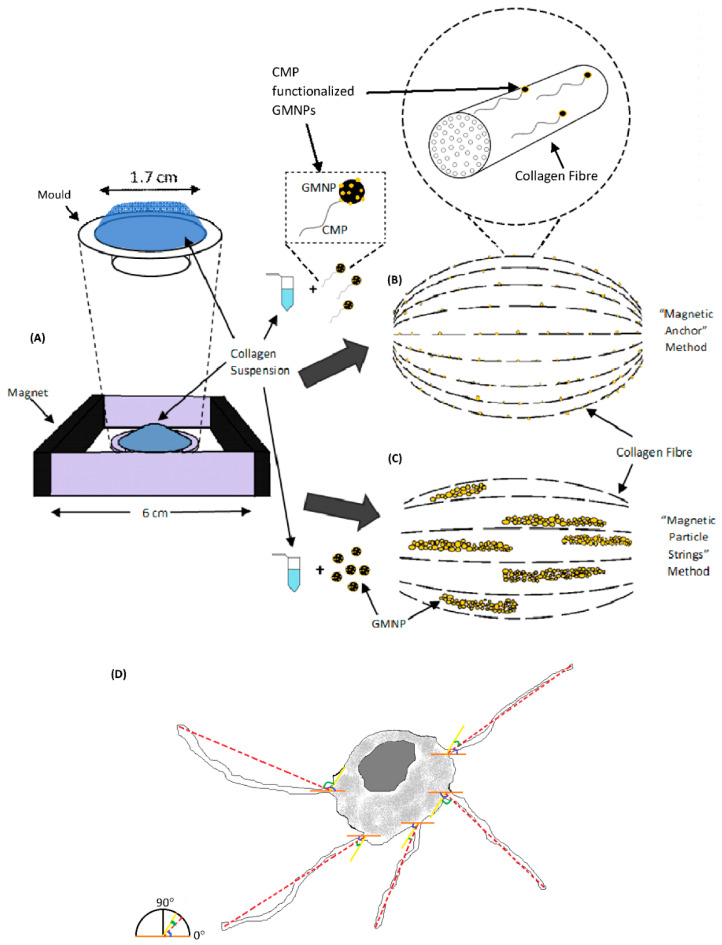
Schematic overview of the magnetic field set-up and methods compared in aligning collagen fibres (**A**). The experimental set-up that enabled the application of magnetic field to collagen gels (**B**). Collagen fibre alignment using the “magnetic anchor” method (**C**). The PC12 cell schematic in (**D**) shows the method by which the angle of alignment (blue angle) and the angle of deviation from the mean angle of alignment (green angle) was derived. The dotted red line represents the neurites’ alignment plane. The orange line represents the horizontal plane (0°). The yellow line represents the mean angle of the alignment plane.

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
