# Peer review of "Application of “Magnetic Anchors” to Align Collagen Fibres for Axonal Guidance"

_gels, 2021, doi:10.3390/gels7040154_

Round 1

Reviewer 1 Report

Title:

  • Consistency: Either use “fibre” (brithish English) or “fiber” consistently throughout the paper (e.g. nanofiber)

Abstract:

  • Please spell out acronyms (e.g., GMNP and CMP) at their first reference

Introduction

  • A few language mistakes: singular/plural, word choice (e.g. ‘communicable’ ND, ‘increment’ of the aged population)

Results and Discussion

  • Magnetic particle string was not introduced in Introduction.
  • Use acronyms for various methods to ease reading (e.g. MPS for magnetic particle string. MA for magnetic anchor, etc.)
  • Fig 1: Highlight important aspects such as aggregations
  • 1. UV Vis:
    • Fig 2: Absorbance should not exceed 100%
  • 2. SEM Image Analysis:
    • Split up average peak height for different method instead of combining all to 0.024
    • Fig 3: Caption (A, B,…, F) with white background to improve visibility; use same font
      • How many samples were analyzed? N=3 required for statistical significance.
    • Suggestion of describing plots in same order as depicted in image (first A, then B, …)
      • Method for 4A, 4B, and 4D not described in text; please provide separate peak values for each method.
    • Fig 4: Add caption with method in each plot for easier visual comparison.
      Include table with numerical values for better comparison of the methods.
  • 3. Confocal Imaging
    • 5C, 5D, and 5E are not associated with respective methods in text.
    • 5: Caption (A, B, …) shifted, missing, partially unreadable
      Add caption with method in each plot for easier visual comparison.
    • ESI Tab 2: How were 25 cells chosen?
    • Splitting methods and results in different sections would help with clarity of the information.
    • 6: Add caption with method in each plot for easier visual comparison.
  • 4. Live/Dead
    • Summary should not be part of this sub-section. Suggestion: Move it to the discussion sub-section
  • 5. Discussion
    • Confirm with journal layout if this should be a subsection or a chapter of its own.
    • Description of “magnetic particle strings” method belongs at the beginning with the other method descriptions.
    • Reference for Antmann-Passig and Orit Shefi 2016 missing
    • Concentration, etc. should be covered in Methods section not at the end of the paper.
    • Split last paragraph into smaller section to ease readability

Materials and Methods

  • Confirm with journal layout if this chapter should come at the end or after the introduction.
  • 3. Reference for Gibco’s collagen gelling protocol missing
  • 8: Caption (A, B, …) shifted, missing, partially unreadable

Author Response

First and foremost, I would like to thank you for taking the time to read and review this study. Your comments are very much appreciated and I have made as much changes as I could to accommodate your remarks. The details of the improvements that I have made per your requests are listed below:

  1. I have changed all the word ‘fibre” to “fibre” and made it consistent throughout the paper.
  2. I have spelled out the acronyms at their first appearance which is in the abstract
  3. Changes have been made to the UV Vis graph. They only show results under 100% absorbance.
  4. I have changed the captions in figure 3 to a white colour for improve visibility.
  5. The samples used in the SEM analysis are three scaffolds per method of fabrication (N=3). I have put them in brackets in the results and discussion section and material and methods section for better clarity as to the number of samples I used. The consistency study in figure 4 shows all the scaffolds in question as well as the reliability of the method.
  6. In image 5, I have moved the A, B…….E labelling in the description section to the end of each statement as to give better and more clarity to the readers.
  7. The summary in the live/dead section has been moved to the discussion section.
  8. I have added a missing reference to the Antmann-Passig and Orit Shefi 2016 paper in the discussion section.
  9. The last part of the discussion has been split in to smaller parts for easier readability.

Reviewer 2 Report

The manuscript "Application of “Magnetic Anchors” to Align Collagen Fibres for Axonal Guidance" is an interesting new approach to achieve injectable orientable scaffolds using magnetic nanoparticles anchored to collagen. By applying a magnetic field, the attraction forces pull the nanoparticles aligining the collagen strings to achieve orientability. The authors present in a clear and convincing manner how the magnetic anchors lead to higher alignment in comparison to non-magnetoresponsive scaffold controls and another method (magnetic strings). As for the manuscript, minor language corrections are needed and the quality of the images require some improvement (see comments below). The following questions arise from and will need to be answered to consider this article complete: 1.- For the creation of the oriented scaffold after injection, this method requires to place the magnets along the axis of orientation, in view of an application in the nerve regeneration field, magnets can only be positioned parallel to the injection site, which would lead to perpendicular orientation of the collagen matrix, how are the authors planing to deal with this issue? 2.- Please elaborate in the discussion on the strenght of the magnetic field required to create a significant force to align the scaffold and how it compares to other methods to create algined injectable scaffolds. 3.- Please elaborate in the discussion any information known about the toxicity and release mechanisms of the nanoparticles in humans and whether they have already been used in medical applicaitons. 4.- Figures: - Figure 1 B and C: images are too dark, and the overlays are all l shifted please fix - Figure 1 C: The particles seem to still aggregate, how does it affect the orientation of the scaffold? 5.- References of importance to be discussed in the manuscript: Incorporating 4D into Bioprinting: Real-Time Magnetically Directed Collagen Fiber Alignment for Generating Complex Multilayered Tissues DOI: 10.1002/adhm.201800894

Author Response

First and foremost, I would like to thank you for taking the time to read and review this study. Your comments are very much appreciated and I have made as much changes as I could to accommodate your remarks. The details of the improvements that I have made per your requests are listed below:

  1. The GMNPs used in this study aligns according to the magnetic field of the magnets used, in whatever shape or direction it might be in. This way the orientation of the collagen fibres in the GMNPs are attached to would also be the same, in regards to the magnetic field. Therefore, technically, if the alignment of neurons within a particular injured site could be determined, then the application would be to inject the gels at the injured site, then finally apply a magnetic field according to the direction of neuronal growth until the collagen has gelled properly. However, further in-depth test needs to be done in applying this new method to align fibres in an in vivo setting.
  2. I have elaborated on the strength of the magnetic field used in this study compared to other methods. The line is as follows; “Additionally, the strength of the magnetic field use in the original magnetic particle string method by Antmann-Passig and Orit Shefi 2016, was 255G for collagen concentration of 3 mg/ml to properly induce alignment (26). In our study the same concentration of collagen was used (3 mg/ml). However, a slightly weaker magnetic force of 64-126G were was used to induce fibre alignment.”
  3. Only a very small part of the scaffolds fabricated with the magnetic anchor method contains aggregation. This may be due to nanoparticles that may not have bound to the CMP and would clump together. However, they will not affect the orientation of the scaffolds in any significant ways as shown in this study with the comparison between the “magnetic nanoparticle string” method and “magnetic anchor” method.
